Optimizing statistical evaluation of multiclass classification in diagnostic radiology: a study of the two-parameter multidimensional nominal response model

Nishio Mizuho 1 jurader@yahoo.co.jp nishiomizuho@gmail.com
Ota Eiji 2
1 Kobe University , Kobe , Japan
2 Futaba Numerical Technologies , Iruma , Japan
Alatas Bilal
Electronic publication date: 2024 Oct 4
Publication date: 2024
Volume: 10
Electronic Location ID: e2380
Received 2024 May 31; Accepted 2024 Sep 10
Copyright: © 2024 Nishio and Ota
Copyright year: 2024
Copyright holder: Nishio and Ota
License: This is an open access article distributed under the terms of the Creative Commons Attribution License, which permits unrestricted use, distribution, reproduction and adaptation in any medium and for any purpose provided that it is properly attributed. For attribution, the original author(s), title, publication source (PeerJ Computer Science) and either DOI or URL of the article must be cited.
License URL: https://creativecommons.org/licenses/by/4.0/

Keywords: Statistics, Nominal response model, Widely applicable information criterion, Pareto-smoothed importance sampling leave-one-out cross-validation, Probability of direction

Funding: JSPS KAKENHI 22K07665 and 23KK0148 Cross-Ministerial Strategic Innovation Promotion Program (SIP) Construction of Integrated Health Care System JPJ01242 This work was supported by JSPS KAKENHI (Grant Numbers: 22K07665 and 23KK0148). This work was also supported by Cross-ministerial Strategic Innovation Promotion Program (SIP) and Construction of Integrated Health Care System (Grant Number: JPJ01242). There was no role of the funding source in the study design, the collection, and analysis and interpretation of data. The funders had no role in study design, data collection and analysis, decision to publish, or preparation of the manuscript.

==============================
Purpose

This study aimed to enhance the multidimensional nominal response model (MDNRM) for multiclass classification in diagnostic radiology.

Materials and Methods

This retrospective study involved the extension of the conventional nominal response model (NRM) to create the two-parameter MDNRM (2PL-MDNRM). Seven models of MDNRM, including the original MDNRM and subtypes of 2PL-MDNRM, were employed to estimate test-takers’ abilities and test item complexity. These models were applied to a clinical diagnostic radiology dataset. Rhat values were calculated to evaluate model convergence. Additionally, values of the widely applicable information criterion (wAIC) and Pareto-smoothed importance sampling leave-one-out cross-validation (LOO) were calculated to evaluate the goodness of fit of the seven models. The best-performing model was selected based on the values of wAIC and LOO. Probability of direction (PD) was used to evaluate whether one estimated parameter significantly differed.

Results

All estimated parameters across the seven models demonstrated Rhat values below 1.10, indicating stable convergence. The best wAIC and LOO values (988 and 1,121, respectively) were achieved with 2PL-MDNRMr using the truncated normal distribution and 2PL-MDNRMa using the truncated normal distribution. Notably, one test-taker (radiologist) exhibited significantly superior ability compared to another based on PD results from the best models, while no significant difference was observed in nonoptimal models.

Conclusion

2PL-MDNRM successfully achieved parameter estimation convergence, and its superiority over the original MDNRM was demonstrated through wAIC and LOO values.

Introduction

Item response theory (IRT) is a statistical framework for constructing, evaluating, and scoring tests and questions to measure test-takers’ capabilities (Hays, Morales & Reise, 2000; Cappelleri, Jason Lundy & Hays, 2014; Gelman et al., 2013). During testing, carefully curated items are presented to evaluate test-takers’ potential. Through IRT, responses provided by test-takers to these items are scrutinized to assess test-takers’ abilities and item complexities. Items used in IRT can be defined for medical diagnosis; for example, test-takers and items of IRT correspond to radiologists and cases in diagnostic radiology, respectively. Consequently, IRT can evaluate the outcomes of observer studies for medical diagnoses. Nishio et al. (2020) used IRT to evaluate radiologists’ ability to detect bone metastasis and brain infarction.

Generally, test-takers’ response data consist of 1 and 0 s for applying IRT, where 1 represents a correct response and 0 represents an incorrect one. However, binary responses do not suit all medical diagnostic scenarios, as multiple types of responses may be involved. For instance, radiologists’ responses in the N factor of cancer TNM grading may include four types (N0, N1, N2, and N3). Consequently, applying IRT to radiologists’ responses in multiclass classification is not straightforward.

The nominal response model (NRM), a multiclass extension of IRT, can be utilized for multiclass classification responses (Carpenter et al., 2017; Luo & Jiao, 2017). Despite reports on Bayesian NRM (Luo & Jiao, 2017), its stability is not guaranteed (The Stan Forums, 2020). Thus, multidimensional NRM (MDNRM) was proposed in a previous study (Nishio et al., 2023). While previous research successfully achieved model convergence using MDNRM (Nishio et al., 2023), its performance as a Bayesian model and statistical evaluation of estimated parameters were not assessed.

This study aimed to enhance the original MDNRM from previous research (Nishio et al., 2023). Hereafter, our improved version of the MDNRM is referred to as the two-parameter MDNRM (2PL-MDNRM). While the original MDNRM was developed from the one-parameter MDNRM (1PL-NRM) in the previous study (Nishio et al., 2023), the current study extended the conventional NRM (2PL-NRM) for 2PL-MDNRM. The development of 2PL-MDNRM was intended to make it possible to efficiently apply MDNRM to medical diagnosis (especially, diagnostic radiology). Generally, the ground truth of multiclass classification is used in the research of medical diagnosis or diagnostic radiology. Therefore, to improve the original MDNRM, the ground truth was considered in 2PL-MDNRM more rigorously than the original MDNRM. To validate the enhancement, values of widely applicable information criterion (Watanabe-Akaike information criterion, wAIC) (Watanabe, 2013, 2010) and Pareto-smoothed importance sampling leave-one-out cross-validation (LOO) (Vehtari et al., 2015) were calculated for the original MDNRM and 2PL-MDNRM. In addition, estimated parameters obtained using 2PL-MDNRM were statistically evaluated. The source code of this study was disclosed as open source through GitHub (https://github.com/jurader/2PL-MDNRM).

Materials and Methods

Dataset

A public dataset from a previous study (Nishio et al., 2022) was used in this study (accessible at https://github.com/jurader/MDNRM/blob/main/ground_truth_and_results.csv). Since a public dataset was used, institutional review board approval or informed consent was not required. The public dataset used in the present study contained classification results obtained from radiologists as described in two previous studies (Nishio et al., 2023, 2022). In addition, the ground truth were included for each item of the dataset. This dataset contained three classes of medical diagnoses: normal, non-novel-coronavirus pneumonia, and novel coronavirus pneumonia, resulting in a 3 × 3 confusion matrix. The dataset consists of 900 entries (150 cases × 6 radiologists), with each nominal response denoted as 0, 1, or 2, representing normal, non-novel coronavirus pneumonia, or novel coronavirus, respectively. Figure 1A shows the representative examples of the ground truth and radiologists’ response. Figure 1B shows the summary of a radiologist’s responses as confusion matrix.

Figure 1 Public dataset used in this study.

(A) The representative examples of the ground truth and radiologists’ response. (B) The summary of one radiologist’s responses as confusion matrix. Note: 0, 1, and 2 represent normal, non-COVID-19 pneumonia, and COVID-19 pneumonia, respectively.

Conventional NRM and 1PL-NRM

Conventional NRM (Luo & Jiao, 2017) serves as a multiclass extension of IRT and is described herein. Additionally, 1PL-NRM (Nishio et al., 2023) is detailed to elucidate the original MDNRM. Conventional NRM (2PL-NRM) extends 2PL-IRT and is represented by the following equations:

(1) Pr(rij=s)=exp⁡(zijs)∑h=1cexp⁡(zijh)zijs=αisθj+βis

where Pr(rij=s) represents the probability that the response of test-taker j to item i is class s,

The number of classes is c,

αis and βis are the two parameters of item i on class s (discrimination and easiness parameters),

θj represents the ability parameter of test-taker j.

In NRM, the softmax function is used to convert logit ( zijs) to probability. Based on the results of the multiclass classification, the parameters of NRM ( αis,βis,andθj in conventional NRM) are estimated.

Previously, 1PL-NRM was employed to stabilize the Bayesian NRM results (Nishio et al., 2023). In 1PL-NRM, the discrimination parameter (α) is removed from conventional NRM. The following equations represent 1PL-NRM.

(2) Pr(rij=s)=exp⁡(zijs)∑h=1cexp⁡(zijh)zijs=θj+βis.

Original MDNRM

The original MDNRM was proposed as an extension of 1PL-NRM (Nishio et al., 2023). The following equations represent the original MDNRM. The ability parameter of test-taker ( θjst) of 1PL-NRM is a multidimensional matrix in the original MDNRM.

(3) Pr(rij=t|groundtruthofitemi=s)=exp⁡(zijst)∑h=1cexp⁡(zijsh)zijst=θjst+βis

where s represents the ground truth of case i,

θjst represents the ability parameter of test-taker j in class t when the ground truth of the item is s.

2PL-MDNRM

The conventional NRM (2PL-NRM) was extended to 2PL-MDNRM. The ability parameter of test-taker ( θ) in 2PL-NRM becomes a multidimensional matrix in 2PL-MDNRM. In addition, the discrimination and easiness parameters ( α and β) can be multidimensional vectors. The following equations represent the three subtypes of the 2PL-MDNRM:

(4) Pr(rij=t|groundtruthofitemi=s)=exp⁡(zijst)∑h=1cexp⁡(zijsh)zijst=αisθjst+βs

(5) Pr(rij=t|groundtruthofitemi=s)=exp⁡(zijst)∑h=1c⁡exp⁡(zijsh)zijst=αsθjst+βis

(6) Pr(rij=t|groundtruthofitemi=s)=exp⁡(zijst)∑h=1cexp⁡(zijsh)zijst=αisθjst+βis.

The three subtypes of 2PL-MDNLM shown in Eqs. 4–6 are referred to as 2PL-MDNRMa, 2PL-MDNRMb, and 2PL-MDNRMr. In these subtypes, the discrimination parameter ( α), easiness parameter ( β), and both discrimination and easiness parameters ( α and β) are multidimensional. The relationship between 1PL-NRM, 2PL-NRM, original MDNRM, and 2PL-MDNRM is shown in Fig. 2.

Figure 2 Relationship between 2PL-NRM (conventional NRM), 1PL-NRM, original MDNRM, and 2PL-MDNRM.

Abbreviations: NRM, nominal response model; MDNRM, multidimensional nominal response model.

Experiments

In this study, the original MDNRM and 2PL-MDNRM were applied to the public dataset to analyze the abilities of six radiologists across three classes. The following prior distributions were applied: For all ability ( θ) and easiness ( β) parameters, normal distribution with mean = 0 and standard deviation = 2 was used.

For discrimination parameters (α), gamma distribution (alpha = 2 and beta = 2) or truncated normal distribution (a half-normal distribution with scale = 10) were used.

Parameters were estimated from the 900 nominal responses of the public dataset for the following seven models: (i) original MDNRM with gamma distribution, (ii) 2PL-MDNRMa with gamma distribution, (iii) 2PL-MDNRMa with truncated normal distribution, (iv) 2PL-MDNRMb with gamma distribution, (v) 2PL-MDNRMb with truncated normal distribution, (vi) 2PL-MDNRMr with gamma distribution, and (vii) 2PL-MDNRMr with truncated normal distribution.

NumPyro was used to implement the seven MDNRM models (Nishio et al., 2023; Phan, Pradhan & Jankowiak, 2019). The following parameters were used for sampling in NumPyro: chains = 8, number of sampling = 8,000, number of warmup = 1,000. The following software packages were used to implement the MDNRM: Python (version 3.10.12; Python Software Foundation, Wilmington, DE, USA), NumPyro (version 0.10.1), Jax (version 0.4.23), and ArviZ (version 0.15.1).

Evaluation

A convergence check of the original MDNRM and 2PL-MDNRM was performed by evaluating the Rhat values of all parameters. If the Rhat values of all parameters were less than 1.10, it was assumed that the Bayesian model had converged stably. To evaluate the goodness of fit of the Bayesian model, the wAIC values were calculated using the ArviZ package. Additionally, the LOO values were calculated. Based on the wAIC and LOO values, the best-performing model was selected from the seven ((i)–(vii)) (Watanabe, 2013, 2010; Vehtari et al., 2015; Luo & Al-Harbi, 2017). The probability of direction (PD) was used to evaluate whether one estimated parameter significantly differed from another (Makowski et al., 2019; Makowski, Ben-Shachar & Lüdecke, 2019). We adopted a 94% highest density interval for PD as it is the default setting in the ArviZ package.

Results

From the Eqs. 3–6, the parameter dimensions of MDNRM subtypes are summarized as follows.

Original MDNRM

θjst: (number of test-takers) × (number of classes) × (number of classes)

βis: (number of items) × (number of classes)

2PL-MDNRMa

θjst: (number of test-takers) × (number of classes) × (number of classes)

αis: (number of items) × (number of classes)

βs: (number of classes)

2PL-MDNRMb

θjst: (number of test-takers) × (number of classes) × (number of classes)

αs: (number of classes)

βis: (number of items) × (number of classes)

2PL-MDNRMr

θjst: (number of test-takers) × (number of classes) × (number of classes)

αis: (number of items) × (number of classes)

βis: (number of items) × (number of classes)

The seven models of the original MDNRM and 2PL-MDNRM were applied to the public dataset to analyze the abilities of six radiologists across three classes. Parameters were estimated from the 900 nominal responses of the public dataset for each model.

All estimated parameters in each of the seven models had Rhat values of less than 1.10, indicating stable convergence as Bayesian models. In fact, the Rhat values are close to 1.00 in the optimal model, indicating good model convergence. This was a major contrast between the Bayesian NRM in the previous study (Luo & Jiao, 2017; The Stan Forums, 2020) and the seven models in the present study. Table 1 and Figs. 3 and 4 present the wAIC and LOO values for these models, with ranges of 988–1,176 and 1,121–1,207, respectively. Based on these values, the truncated normal distribution outperforms the gamma distribution for 2PL-MDNRMa, 2PL-MDNRMb, and 2PL-MDNRMr. Notably, the smallest wAIC and LOO values (988 and 1,121, respectively) were observed in 2PL-MDNRMr with the truncated normal distribution and 2PL-MDNRMa with truncated normal distributions. These two models were identified the best-performing models in the present study.

Table 1 Values of wAIC and LOO in the seven models of MDNRM.

Model	wAIC	LOO	
(i) Original MDNRM	1,147 ± 47.8	1,177 ± 49.6	
(ii) 2PL-MDNRMa with the gamma distribution	1,175 ± 40.6	1,207 ± 42.4	
(iii) 2PL-MDNRMa with the truncated normal distribution	999 ± 40.8	1,121 ± 48.1	
(iv) 2PL-MDNRMb with the gamma distribution	1,145 ± 47.6	1,174 ± 49.3	
(v) 2PL-MDNRMb with the truncated normal distribution	1,146 ± 47.7	1,175 ± 49.4	
(vi) 2PL-MDNRMr with the gamma distribution	1,096 ± 43.5	1,169 ± 47.1	
(vii) 2PL-MDNRMr with the truncated normal distribution	988 ± 43.4	1,165 ± 52.2	
Note:

Because there is no discrimination parameter in original MDNRM, it is impossible to use original MDNRM with the truncated normal distribution. Abbreviations: wAIC, widely applicable information criterion; LOO, Pareto-smoothed importance sampling leave-one-out cross-validation; MDNRM, multidimensional nominal response model.

Figure 3 Results of wAIC in the seven models of MDNRM.

Abbreviations: wAIC, Watanabe-Akaike information criterion; MDNRM, multidimensional nominal response model.

Figure 4 Results of LOO in the seven models of MDNRM.

Abbreviations: LOO, Pareto-smoothed importance sampling leave-one-out cross-validation; MDNRM, multidimensional nominal response model.

Subsequent to this, the estimation results for the easiness and discrimination parameters are omitted as our focus is on the ability parameters. For the best-performing models identified (2PL-MDNRMr with a truncated normal distribution in wAIC and 2PL-MDNRMa with a truncated normal distribution in LOO), Tables 2 and 3 present the estimation results of the ability parameters for the six radiologists. Each radiologist’s ability is represented by a 3 × 3 matrix in the MDNRM. Higher diagonal values in the ability parameter matrix indicate higher test-taker’s abilities. Tables 2 and 3 include the estimation results for the 54 ability parameters (6 × 3 × 3).

Table 2 Estimation results of ability parameters in MDNRMr.

rad_index	index1	index2	Mean	SD	HDI_3%	HDI_97%	Rhat	
0	0	0	0.177	0.22	−0.202	0.579	1.00	
0	0	1	−0.505	0.375	−1.187	0.115	1.00	
0	0	2	−1.481	0.888	−3.122	−0.118	1.00	
0	1	0	−1.118	0.957	−2.991	0.202	1.00	
0	1	1	0.255	0.366	−0.335	0.97	1.00	
0	1	2	0.052	0.31	−0.468	0.636	1.00	
0	2	0	−0.967	0.59	−2.075	−0.136	1.00	
0	2	1	−0.328	0.197	−0.707	0.005	1.00	
0	2	2	−0.04	0.139	−0.31	0.213	1.00	
1	0	0	0.982	0.904	−0.033	2.854	1.00	
1	0	1	−0.045	0.437	−0.798	0.871	1.00	
1	0	2	−0.031	0.37	−0.651	0.705	1.00	
1	1	0	−0.98	0.83	−2.634	0.025	1.00	
1	1	1	0.114	0.29	−0.495	0.628	1.00	
1	1	2	−0.332	0.458	−1.179	0.216	1.00	
1	2	0	−1.606	0.879	−3.239	−0.263	1.00	
1	2	1	−0.314	0.194	−0.676	0.03	1.00	
1	2	2	0.164	0.166	−0.141	0.48	1.00	
2	0	0	0.358	0.306	−0.165	0.936	1.00	
2	0	1	−0.534	0.564	−1.558	0.189	1.00	
2	0	2	−1.09	0.743	−2.503	−0.039	1.00	
2	1	0	−1.771	1.231	−4.17	−0.025	1.00	
2	1	1	−0.002	0.379	−0.821	0.648	1.00	
2	1	2	−0.647	0.729	−2.203	0.29	1.00	
2	2	0	−0.907	0.677	−2.195	0.011	1.00	
2	2	1	−0.011	0.158	−0.314	0.287	1.00	
2	2	2	0.491	0.27	0.053	0.993	1.00	
3	0	0	1.645	1.098	0.105	3.78	1.00	
3	0	1	−0.413	0.629	−1.654	0.538	1.00	
3	0	2	−2.091	1.208	−4.326	−0.094	1.00	
3	1	0	−0.502	0.277	−1.009	−0.033	1.00	
3	1	1	−0.081	0.15	−0.361	0.204	1.00	
3	1	2	−0.218	0.169	−0.541	0.086	1.00	
3	2	0	−0.763	0.367	−1.447	−0.216	1.00	
3	2	1	−2.712	1.037	−4.665	−0.992	1.00	
3	2	2	−0.477	0.248	−0.94	−0.084	1.00	
4	0	0	−0.247	0.321	−0.831	0.318	1.00	
4	0	1	−1.523	1.041	−3.437	0.053	1.00	
4	0	2	−2.279	1.201	−4.46	−0.25	1.00	
4	1	0	−1.607	0.923	−3.386	−0.217	1.00	
4	1	1	−0.14	0.211	−0.531	0.225	1.00	
4	1	2	−0.213	0.232	−0.628	0.185	1.00	
4	2	0	−1.091	0.679	−2.383	−0.173	1.00	
4	2	1	−0.522	0.304	−1.082	−0.053	1.00	
4	2	2	−0.267	0.218	−0.683	0.07	1.00	
5	0	0	2.378	1.31	0.27	4.777	1.00	
5	0	1	−0.609	0.883	−2.439	0.585	1.00	
5	0	2	−1.959	1.21	−4.221	−0.014	1.00	
5	1	0	−0.696	0.729	−2.172	0.156	1.00	
5	1	1	−0.674	0.73	−2.147	0.188	1.00	
5	1	2	−0.847	0.871	−2.63	0.129	1.00	
5	2	0	−1.295	0.591	−2.394	−0.328	1.00	
5	2	1	−2.546	1.025	−4.472	−0.822	1.00	
5	2	2	−1.435	0.646	−2.65	−0.398	1.00	
Note:

For example, the row for rad_index = 1, index1 = 2, and index2 = 2 represents the ability parameter of θ122.

Table 3 Estimation results of ability parameters in MDNRMa.

rad_index	index1	index2	Mean	SD	HDI_3%	HDI_97%	Rhat	
0	0	0	−0.024	0.155	−0.317	0.263	1.00	
0	0	1	−0.678	0.436	−1.464	−0.05	1.00	
0	0	2	−1.472	1.01	−3.385	−0.034	1.00	
0	1	0	−1.694	0.802	−3.2	−0.443	1.00	
0	1	1	−0.202	0.086	−0.367	−0.051	1.00	
0	1	2	−0.125	0.104	−0.322	0.054	1.00	
0	2	0	−1.164	0.603	−2.287	−0.278	1.00	
0	2	1	−0.529	0.23	−0.953	−0.157	1.00	
0	2	2	−0.033	0.114	−0.248	0.176	1.00	
1	0	0	0.206	0.237	−0.121	0.628	1.00	
1	0	1	−0.432	0.373	−1.106	0.094	1.00	
1	0	2	−0.086	0.219	−0.474	0.239	1.00	
1	1	0	−1.524	0.748	−2.912	−0.378	1.00	
1	1	1	−0.265	0.138	−0.513	−0.05	1.00	
1	1	2	−0.45	0.333	−1.02	−0.018	1.00	
1	2	0	−1.849	0.961	−3.666	−0.361	1.00	
1	2	1	−0.405	0.182	−0.748	−0.105	1.00	
1	2	2	0.16	0.128	−0.065	0.413	1.00	
2	0	0	0.065	0.212	−0.322	0.471	1.00	
2	0	1	−0.83	0.679	−2.106	0.017	1.00	
2	0	2	−0.916	0.758	−2.343	0.064	1.00	
2	1	0	−1.396	0.703	−2.697	−0.34	1.00	
2	1	1	−0.14	0.096	−0.316	0.018	1.00	
2	1	2	−0.254	0.215	−0.599	0.038	1.00	
2	2	0	−1.23	0.763	−2.662	−0.155	1.00	
2	2	1	−0.206	0.129	−0.449	0.022	1.00	
2	2	2	0.325	0.184	0.02	0.671	1.00	
3	0	0	1.367	1.112	−0.128	3.487	1.00	
3	0	1	−0.478	0.66	−1.787	0.464	1.00	
3	0	2	−1.728	1.227	−3.982	0.136	1.00	
3	1	0	−2.167	1.03	−4.068	−0.528	1.00	
3	1	1	−1.154	0.593	−2.25	−0.253	1.00	
3	1	2	−1.319	0.795	−2.806	−0.156	1.00	
3	2	0	−0.892	0.4	−1.629	−0.287	1.00	
3	2	1	−2.861	1.061	−4.841	−1.085	1.00	
3	2	2	−0.421	0.271	−0.927	−0.015	1.00	
4	0	0	−0.379	0.247	−0.857	0.041	1.00	
4	0	1	−1.82	0.984	−3.61	−0.226	1.00	
4	0	2	−2.084	1.191	−4.213	−0.14	1.00	
4	1	0	−1.768	0.771	−3.217	−0.551	1.00	
4	1	1	−0.31	0.125	−0.546	−0.106	1.00	
4	1	2	−0.182	0.132	−0.424	0.036	1.00	
4	2	0	−0.799	0.382	−1.5	−0.244	1.00	
4	2	1	−0.456	0.193	−0.803	−0.161	1.00	
4	2	2	−0.079	0.118	−0.29	0.115	1.00	
5	0	0	2.075	1.292	0.029	4.408	1.00	
5	0	1	−0.461	0.735	−1.955	0.536	1.00	
5	0	2	−1.649	1.24	−3.978	0.204	1.00	
5	1	0	−0.492	0.334	−1.028	−0.102	1.00	
5	1	1	−0.556	0.363	−1.124	−0.134	1.00	
5	1	2	−0.402	0.401	−1.015	0.027	1.00	
5	2	0	−1.384	0.62	−2.504	−0.346	1.00	
5	2	1	−2.657	1.07	−4.623	−0.81	1.00	
5	2	2	−1.397	0.725	−2.726	−0.229	1.00	
Note:

For example, the row for rad_index = 1, index1 = 2, and index2 = 2 represents the ability parameter of θ122.

As representative examples, the ability parameters for novel coronavirus pneumonia were compared using PD between radiologists 1 and 5 (between θ122 and θ522). Figures 5–7 depict the representative PD results for the three MDNRM models. θ122 was significantly better than θ522 for the two best-performing models: 2PL-MDNRMr with a truncated normal distribution and 2PL-MDNRMa with a truncated normal distribution. Conversely, θ122 was not significantly better than θ522 for the original MDNRM. Given that the wAIC and LOO values of the original MDNRM were not as favorable as those of the two best-performing models, statistical significance was not attained for the original MDNRM.

Figure 5 Results of posterior distribution and PD in original MDNRM.

Note: The upper row displays the posterior distribution of ability parameters of radiologists 1 and 5 (θ122 and θ522). The lower row presents the PD results for ability parameters. Abbreviations: PD, probability of direction; MDNRM, multidimensional nominal response model.

Figure 6 Results of posterior distribution and PD in 2PL-MDNRMr.

Note: The upper row displays the posterior distribution of ability parameters for radiologists 1 and 5 (θ122 and θ522). The lower row shows the PD results for ability parameters. Abbreviations: PD, probability of direction; MDNRM, multidimensional nominal response model.

Figure 7 Results of posterior distribution and PD in MDNRMa.

Note: The upper row shows the posterior distribution of ability parameters of radiologists 1 and 5 (θ122 and θ522). The lower row shows the PD results for ability parameters. Abbreviation: PD, probability of direction; MDNRM, multidimensional nominal response model.

Discussion

Our study focused on addressing the stability of convergence and comparing the performance of different models in the context of multiclass classification of diagnostic radiology using the MDNRM. We found that all seven models demonstrated successful convergence, as indicated by Rhat values consistently below 1.10. Notably, the models of 2PL-MDNRMr with the truncated normal distribution and 2PL-MDNRMa with truncated normal distribution emerged as the top performers, achieving the smallest wAIC and LOO values, respectively. The PD results further confirmed the feasibility of statistically comparing ability parameters in these two best-performing models.

Given the previous instability of convergence in the conventional NRM (The Stan Forums, 2020), the significance of multidimensional ability parameters for stable convergence in MDNRM is suggested in the previous study (Nishio et al., 2023) and the current study. In addition, our results showed that the Bayesian models’ goodness of fit (wAIC and LOO values) was influenced by parameter dimensions and the use of a truncated normal distribution. The preference for the truncated normal distribution over the gamma distribution for the prior distribution of discrimination parameters is supported by previous studies (Röver et al., 2021; Gelman, 2006), hence the truncated normal distribution was adopted in the present study. Our results show that the parameter dimensions were important for improving the wAIC and LOO values, which means that the original MDNRM cannot adequately grasp the data generation process of multiclass classification.

In the original NDNRM and 2PL-NDNRM, the test-takers’ abilities were represented as a matrix. The diagonal values of this matrix are crucial in MDNRM; higher diagonal values indicate greater test-takers’ ability. Conversely, lower easiness parameter values indicate more challenging items. The discrimination parameters in MDNRM are similar to those in 2PL-IRT (Hays, Morales & Reise, 2000; Cappelleri, Jason Lundy & Hays, 2014). Therefore, the discrimination parameters can be utilized to judge whether the test items can be used to easily evaluate test-takers (radiologists) (Cappelleri, Jason Lundy & Hays, 2014; Nishio et al., 2023).

Despite achieving stable convergence in the original MDNRM, previous studies did not statistically compare ability parameters. Our results demonstrate the feasibility of such a comparison using PD and the optimal 2PL-MDNRM. This combination of PD and 2PL-MDNRM may offer a useful alternative to conventional statistical tests for evaluating estimated parameters.

While the default value of “94%” for PD in ArviZ packages was employed in this study, careful consideration must be given to its selection. Although “p = 0.05” is commonly used as the significance threshold in conventional statistical tests, it is not absolute, especially in cases of multiple testing where adjustments such as the Bonferroni or Holm corrections are necessary. Similarly, “94%” for PD must be carefully chosen based on the characteristics of studies. However, if the threshold of PD is carefully chosen, PD results in Bayesian models can be used instead of p-value in conventional statistical test.

Conventional models of latent response theory, such as item response theory or multidimensional nominal response model, have been used in clinical and psychometric studies (Zhao et al., 2023; Revuelta & Ximénez, 2017; Falk & Ju, 2020). These studies mainly focus on the response obtained from the experiment where the ground truth is not available or defined. On the other hand, the development of 2PL-MDNRM is intended to be used for the studies where the ground truth is available. This point is a major difference between conventional models and 2PL-MDNRM. Another major difference is the dimensionality of the ability parameters. Although D-dimensional parameters (D-dimensional vector) were used in the conventional multidimensional nominal response model, CxC-dimensional parameters (CxC-dimensional matrix) were used in 2PL-MDNRM. In addition, although the constrained versions of the multidimensional nominal response model were used in psychometric studies, the constraint was not necessary in 2PL-MDNRM.

This study had some limitations. First, although 900 nominal responses were utilized, the minimum number required for stable convergence remains unclear. Second, statistical significance depends on the models’ goodness or power, but this study did not elucidate how the goodness of fit influences the statistical significance of PD. Third, only one dataset was used in this study. As a result, it was not possible to evaluate whether the results of the current study could be reproduced in other datasets. Evaluating the generalizability of 2PL-MDNRM is a future study. Fourth, we did not evaluate multiple types of prior distributions. For example, use of the standard normal distribution was not evaluated in this study. Because the source code of this study is publicly available, we hope that the effect of the prior distribution will be evaluated rigorously in a future study. Fifth, multivariate normal distribution was not used as prior distribution for multidimensional latent traits. Sixth, although three types of 2PL-MDNRM were suggested in the current study, other variants of MDNRM can be suggested. For example, the ability parameter can be a multidimensional vector instead of a multidimensional matrix. We believe that the usefulness of other variants can be evaluated using the wAIC or LOO values.

Conclusions

The proposed 2PL-MDNRM offers a robust solution for multiclass classification in diagnostic radiology. Achieving successful convergence in parameter estimation akin to other Bayesian models, the superiority of 2PL-MDNRM over the original MDNRM is evident from the wAIC and LOO values. Our results also shows that the use of wAIC and LOO values may be useful for selecting Bayesian models. In addition, the feasibility of comparing ability parameters using PD underscores its potential utility in this context.

Additional Information and Declarations

Competing Interests

Author Contributions

Data Availability

Eiji Ota is employed by Futaba Numerical Technologies. The authors declare that they have no other competing interests.

Mizuho Nishio conceived and designed the experiments, performed the experiments, analyzed the data, performed the computation work, prepared figures and/or tables, authored or reviewed drafts of the article, and approved the final draft.

Eiji Ota performed the experiments, analyzed the data, performed the computation work, authored or reviewed drafts of the article, and approved the final draft.

The following information was supplied regarding data availability:

The code and raw data are available at GitHub and Zenodo:

- https://github.com/jurader/2PL-MDNRM.

- jurader. (2024). jurader/2PL-MDNRM: 1st release (second). Zenodo. https://doi.org/10.5281/zenodo.13751068.

The third-party data is available at: doi: https://doi.org/10.1007/S11604-022-01366-Y.

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
