# Peer review of "Optimizing statistical evaluation of multiclass classification in diagnostic radiology: a study of the two-parameter multidimensional nominal response model"

_PeerJ Computer Science, doi:10.7717/peerj-cs.2380_

## Round 0.1 · original submission · Major Revisions

Dear authors,

Reviewers' feedback is now available. It is not recommended that your article be published in its current format. However, we strongly recommend that you address the issues pointed out by the reviewers, especially those concerning readability, experimental design and validity, and resubmit your paper after making the necessary changes.

Best wishes,

Reviewer 1 ·

Basic reporting

The manuscript presents a study on the optimization of statistical evaluation methods for multiclass classification in diagnostic radiology, focusing on the two-parameter multidimensional nominal response model (2PL-MDNRM). The authors aim to enhance the existing MDNRM by incorporating two-parameter models, evaluate the performance of different models, and compare the effectiveness of various statistical measures.

Strengths:

Novelty and Relevance: The study addresses a pertinent issue in diagnostic radiology, specifically in the application of item response theory (IRT) to multiclass classification. The proposed 2PL-MDNRM is a significant contribution as it extends the traditional NRM and offers a robust solution for analyzing radiologists' performance.

Methodological Rigor: The use of advanced statistical techniques, including Bayesian modeling, Rhat values for convergence checking, and model comparison metrics like wAIC and LOO, demonstrates a thorough and rigorous approach.

Comprehensive Analysis: The manuscript compares different model specifications and discusses the implications of using various prior distributions (gamma and truncated normal). This thorough analysis helps in understanding the strengths and limitations of each model.

Clarity and Structure: The manuscript is well-organized, with clear sections for the introduction, methodology, results, and discussion. The inclusion of equations and detailed explanations of the models used is helpful for readers.

Areas for Improvement:

While the introduction provides a good overview of IRT and its relevance, it could benefit from a more detailed explanation of the specific challenges faced in multiclass classification in radiology. This would help to better contextualize the need for the proposed 2PL-MDNRM. The introduction and problem statement is rather too scanty.
The literature review could be expanded to include more recent studies on IRT applications in medical diagnostics and highlight more existing gaps that this study aims to fill.

Experimental design

The results section is comprehensive, but the presentation of findings could be enhanced with more visual aids, such as graphs or tables summarizing the key comparison metrics (wAIC, LOO, PD) across models. This would make it easier for readers to quickly grasp the performance differences.
The discussion of the PD results could be expanded to explain the practical implications of the findings, particularly how they could impact clinical practice or further research in radiology.

Validity of the findings

The Rhat values are all listed as 1.00, which suggests good convergence of the model. This could be briefly mentioned in the results or discussion section to reassure readers of the reliability of the estimates.
Consistency and Accuracy:

Double-check the numbers and calculations for consistency and accuracy. Ensure that all values align with the data and methodology used.

Additional comments

The discussion should delve deeper into the limitations of the study, particularly regarding the use of a specific dataset and how generalizable the findings are to other medical contexts or datasets.
The conclusion could provide more actionable recommendations for practitioners and researchers, such as best practices for selecting statistical models in similar studies.

While the technical explanations are generally clear, some parts (e.g., the description of prior distributions) could be simplified or explained in more layman's terms to ensure accessibility to a broader audience.
Consider including more details in the discussion and conclusion sections.

Reviewer 2 ·

Basic reporting

Although the authors "extended the conventional NRM" to 2PL-MDNRM, the proposed model is not new. Multidimensional nominal response models have been proposed and widely used in psychometric literature (see, e.g., Revuelta & Ximénez, Frontiers in Psychology, 2017). Therefore, this study is a pure simulation study that compares different models. Previous studies on MDNRM should be cited.

Experimental design

1. The way the authors model the multidimensional theta is unusual and different from previous studies. Since only the ability to correctly identify classes is important, only three dimensions are needed in the model. For example, it is more common to assume that Pr(r_ij=t|ground truth of item i=s) only depends on theta_js.
2. If the ground truth of item i is s, then only beta_is is needed, while beta_it where t != s never appears in the model. As a result, the number of betas should be equal to the number of items.
3. Since z = alpha * theta + beta rather than z = alpha * theta - beta, beta should be interpreted as an easiness rather than a difficulty parameter.
4. In Equation (4), when the slopes can be different, it is unreasonable that the intercepts can stay the same. Therefore, Model (4) is not useful, while keeping Models (5) and (6) is sufficient.
5. The metric of theta is almost always set as N(0,1) in IRT analysis. Is there any specific reason why N(0,2^2) is used as a prior for theta? Also, in multidimensional IRT models, different dimensions should be correlated, otherwise we can simply run several unidimensional models. Therefore, the prior for theta should be a multivariate normal.

Validity of the findings

no comment

Additional comments

The goal of the study is not well-stated. According to the current manuscript, it is a simulation study that compares different model specifications.

---

## Round 0.2 · accepted · Accept

Dear authors,

Thank you for clearly performing the necessary additions and modifications. The paper now seems ready for publication.

Best wishes,

Reviewer 2 ·

Basic reporting

No comment

Experimental design

No comment

Validity of the findings

No comment